# Intravital Position Study of the Clinical Anatomy of the Middle Lobe and Superior Poles of the Thyroid Gland

**DOI:** 10.3390/medicina60091520

**Published:** 2024-09-18

**Authors:** Vladislav V. Tatarkin, Andrey I. Shchegolev, Aleksandr M. Bakunov, Yuriy L. Vasil’ev, Mikhail D. Chernyshev, Evgeniy A. Serebryakov, Ksenia L. Kaplieva, Egor O. Stetsik, Tahmina Pulotova, Ellina V. Velichko, Evgeniy M. Trunin

**Affiliations:** 1Department of Operative and Clinical Surgery with Topographical Anatomy, I. I. Mechnikov Northwestern State Medical University of the Ministry of Health of the Russian Federation, 191015 Saint Petersburg, Russia; vlad1k2@ya.ru (V.V.T.); cyonic@yandex.ru (A.I.S.); mih.chernyshyov@yandex.ru (M.D.C.); evgeniyserebrjakov@yandex.ru (E.A.S.); k.l.kaplieva2002@gmail.com (K.L.K.); egorst2564@mail.ru (E.O.S.); t_pulatova01@mail.ru (T.P.); etrunin@mail.ru (E.M.T.); 2Saint-Petersburg State Medical Institution, Elizavetinskaya Hospital, 195427 Saint Petersburg, Russia; sanka-86@bk.ru; 3Department of Operative Surgery and Topographic Anatomy, Sechenov First Moscow State Medical University of the Ministry of Health of the Russian Federation (Sechenov University), 119048 Moscow, Russia; velichko_e_v@staff.sechenov.ru

**Keywords:** positional anatomy, thyroid lobe, thyroid gland, neck MRI

## Abstract

*Background and Objectives*: This study analyzes the issues of position dislocation of the head of the superior poles and lateral points of the outer edge of the middle divisions of each of the thyroid lobes (TL) changes. The purpose was to provide an intravital position study of the clinical anatomy of the middle and superior poles of the thyroid gland (TG). *Materials and Methods*: We used data on anthropometry obtained during the study and data from MRI of the neck organs and anatomical formations of 100 healthy individuals, comprising 55 (55%) female and 45 (45%) male participants. To evaluate the data obtained in the examined group, the minimum probability value was 0.95 (95% confidence interval or *p* < 0.05). *Results*: Maximum flexion of the neck the distance from the center of the jugular notch to the lateral points of the outer edge of the middle parts of the TL decreases in all groups of but mostly in male ectomorphs; the same distance to the superior poles of the TG changes most in ectomorphic males on the right by 38.9% (*p* value < 0.001) and mesomorphic females on the left by 37.8% (*p* value < 0.001); in rotation to the left, the maximum was found in male ectomorphs, which decreased by 9.5% on the left (*p* value < 0.001) and by 7.3% on the right (*p* value < 0.001). *Conclusions*: this study provided new information about the intravital anatomy of the TG position, of the superior poles, and the lateral points of the middle part of its lobes.

## 1. Introduction

Positional anatomy is a budding areas in modern medicine. It aims to study the relative positions of organs and systems when the normal position of the body is changed (e.g., flexion/extension, abduction/adduction, rotations/circumductium, supination/pronation, eversion/inversion retraction/protraction, etc.). It is difficult to find a body part more complicated by its structure area, with a larger number of vital structures and with less accessible area for movements of instruments during the operation, than the neck. It combines the most important anatomical objects, organs, and systems in the body, such as the spinal cord, cranial nerves, nerve plexuses (central nerve system), common carotid arteries and their main branches, vertebral arteries, internal jugular, and other main veins (cardio vascular systems). thoracic lymphatic ducts, multiple lymph nodes (lymphatic system), larynx, trachea (respiratory system), pharynx, esophagus (digestive system), thyroid and parathyroid glands (endocrine system), cervical spine (skeletal system), large number of muscles that support the head and are involved in turning the head and neck (muscle system), and the skin (integumentary system) [1].

The human neck is biomechanically unique in terms of its mobility. Neck can be flexed at 85° and extended at 70°. With a head rotating, the neck is twisted to the right and left by 80% [2,3]. The relative position of the organs and anatomical formations of the neck change when the head position is changed, i.e., it rotates left/right, extended, flexed) [4]. Currently, minimally invasive surgical access is widely used for surgical neck interventions, including thyroid and parathyroid glands [5,6,7]. Intraoperative changes in the position of the head and neck contribute to the fact that the areas of the thyroid lobes subjected to surgical manipulations become closer to or farther from the skin wound [8]. The information about the intravital positional anatomy of the superior poles of the thyroid gland and lateral points of the middle part of its lobes will help optimize the surgical technique by improving the key topographic and anatomical parameters of the surgical wound.

Our purpose was to gain information on the intravital positional anatomy of the superior poles and lateral areas of the middle parts of each of the thyroid lobes in both males and females with different body shapes.

## 2. Materials and Methods

Anthropometry data and the results of magnetic resonance imaging (MRI) of the neck organs and anatomical structures of 100 healthy volunteers, including 55 (55%) females and 45 (45%) males were used. The median age was 28.48–12.89 years, and the median age was 22 (min. 21; max. 32) years. The maximum age was 65 years, and the minimum age was 18 years. The criteria for inclusion in the study were being older than 18 years, the absence of diseases of the neck organs and surgical interventions on the neck that change its topography in history, and the person’s consent to be included in the study. The work was approved by the Local Ethics Committee of the I. I. Mechnikov Northwestern State Medical University of the Public Health Ministry of the Russian Federation (Protocol No. 9 of 12 December 2022).

Body types were determined using several criteria via the ratio of absolute length of the lower limb to height: if the value was greater than 55, it was considered a dolichomorphic type; if the value was between 55 and 50, it was a mesomorphic type; if it was less than 50, it was a brachymorphic body type [9]. Using the epigastric angle, if it was up to 85°, it was a dolichomorphic body type; from 85° to 95°, it was a mesomorphic type; and more than 95°, it was a brachymorphic type. Using the topographic and anatomical parameters of the neck, the parameter included the length of the neck—i.e., the distance between the chin and superior chest points—and its circumference at the level of the base of the neck, which was used to calculate the width (diameter) of the neck.

The research was conducted using the clinical base of the Department of Operative and Clinical Surgery with Topographic Anatomy named after S. A. Simbirtsev of the Northwestern State Medical University named after I. I. Mechnikov in the St. Petersburg State Budgetary Healthcare Institution City Hospital of the Holy Martyr Elizabeth. Tomography was performed on a specialized two-component coil used as a radio frequency for each of the thyroid lobes. The coil was placed on the skin of the anterior surface of a subject’s neck.

Data analysis was carried out according to the functional study of thyroid gland protocol, and contrast agents were not used (Table 1).

The study was carried out when the subjects were lying on their backs. The direction of neck scanning was caudal: cranial (from bottom to top) in the axial plane, dorsal-ventral (from back to front) in the coronary (frontal) plane, and from right to left in the sagittal plane.

The displacement of the superior poles and lateral parts of the middle parts of each of the thyroid lobes was studied in the standard (physiological) anatomical head and neck position in the position of maximum flexion and extension. Then, the scan was performed with the head rotated to the right and then successively to the left by 45° (Figure 1). In order to fix the subject’s head and neck in non-standard positions, an originally designed device was used [10].

On the obtained MR tomograms, the study of the displacement of paired parts of the thyroid gland was performed bilaterally. In the frontal plane, the location of the maximum superior point of the superior pole and the lateral point of the middle part of the thyroid lobe on each side were determined, and the location of the center of the jugular notch of the sternum was also determined (Figure 2).

To determine the center of the jugular notch of the sternum in the frontal plane, the MRI section was visualized with the maximum depth of the section (mm.). Then, a circle was inscribed on it. The diameter that coincided with the points of the sternoclavicular junction was determined, and a perpendicular caudal radius was plotted from it. The intersection point of the radius and circle was the center of the jugular notch.

To determine the point of the superior pole of the thyroid lobe in the frontal plane, the section was visualized with the maximum distance from the superior to inferior poles of the lobe. Next, an ellipse was inscribed onto the projection of the lobe, after which the diameter connecting the two points most distant in the cranio-caudal direction on the ellipse was plotted. The intersection points of the diameter and circumference of the ellipse, which was the most remote one in the cranial direction, were the highest point of the superior pole of the thyroid lobe. To determine the lateral point of the middle of the lateral surface of the thyroid lobe in the frontal plane, a section was found in which the distance from the outer to the inner edges of the lobe was maximal. Next, an ellipse was inscribed into the projection of the lobe, after which the diameter connecting the two points most distant in the lateral and medial directions on the circle of the ellipse was plotted. The point of intersection of the ellipse diameter with the point of its circumference most distant in the lateral direction was the lateral point of the middle of the thyroid lobe.

To assess the changes in the location of reference points on the thyroid lobes in both males and females with all body types, the distances and changes in distances from the center of the jugular notch to the point of the superior pole of each of the thyroid lobes and the most lateral point of the middle part of each of the thyroid lobes were determined. These distances were measured in the middle anatomical position, with maximum flexing and unflexing of the neck and with rotating the head to the right and left by 45°.

Statistical processing of the obtained results was performed on a personal computer using the following programs: Microsoft Office Excel 2019, Jamovi 1.0.1.9 with the calculation of average values and standard deviations (M ± SD), median and quartiles (Me (LQ; HQ)), minimum and maximum values (min ÷ max), and absolute values and fractions of the whole (%). To evaluate the data obtained in the examined group, the minimum probability value was 0.95 (95% confidence interval or *p* < 0.05).

## 3. Results

In the total sample examined, the neck length was 114 ± 16 mm, the median was 143 (min. 133; max. 154) mm; the neck length was 141 ± 14.2 mm, the median was 140 (min. 130; max. 150) mm for females, ignoring body types; and the neck length was larger and shorter: 154 ± 17.4 mm, with a median of 158 (min. 144; max. 160) mm for males.

When dividing the examined volunteers based on their gender and body type, it was determined that the largest value of the variable neck length occurred in male dolichomorphs. It was 160 mm. The lowest value of this variable was found in males with a brachymorphic body type (130 mm).

In the total sample of the examined group, the neck width (diameter) was 107 ± 9.51 mm, and the median was 108 (min. 100; max. 113) mm. In males, this value was higher—112 ± 9.73 mm and the median was 113 (min. 103; max. 118) mm. The width (diameter) of the neck was 105 ± 9.17 mm, and the median was 105 (min. 110; max. 111) mm in females who were examined by volunteers, without considering the division by body type.

When distributing the examined, considering their gender and body type, it was found that the lowest value of the variable neck width (diameter) occurred in male individuals with a dolichomorphic body type; it was 100 mm. The maximum value of the studied variable was found in male mesomorphic individuals; it was 114 ± 10.2 mm.

The distance in the frontal plane from the center of the jugular notch sternum to the lateral point of the outer edge of the middle part of the right thyroid lobe was determined, considering the sex and body type of the examined. When analyzing the obtained data, it was established that in the average anatomical position, the greatest distance in the frontal plane from the center of the jugular notch of the sternum to the lateral point of the outer edge of the middle part of the right thyroid lobe was found in male endomorphs; it was 82.4 mm. The smallest one was found in female individuals with an endomorphic body type; it was 61.1 mm. Neck flexion reduces the magnitude of this variable in all groups. The maximum value was found in the group of male endomorphs; it was 78.4 mm. The minimum one was found in male ectomorphs (44.1 mm). The greatest decrease in the value of the studied variable was found in the examined males with an ectomorphic body type (42.8%). Neck extension leads to an increase in the distance in the frontal plane from the center of the jugular notch of the sternum to the lateral point of the outer edge of the middle part of the right thyroid lobe: the maximum values of this variable were found in the examined endomorph males; they were 100.1 mm. The minimum values were found in ectomorph males; they were 73.8 mm. The greatest changes occurred in males with an endomorphic physique; it increased by 22%.

Both male and female patients with mesomorphic body types were examined. Rotating a head to the left either leads to a decrease in the studied variable or does not affect its value. The greatest value of the studied distance was found in mesomorph males; it was 75.2 mm. The minimum value was found in the examined endomorph females; it was 53.9 mm. The maximum changes were recorded in the group of examined males with an endomorphic body type; it decreased by 15.9%.

Rotating a head to the right has a multidirectional effect on the value of the studied variable. The examined males with a mesomorphic body type showed an increase in the value of the variable by 1.3%; in all other groups, it decreased. The maximum value of the studied variable was 76.2 mm in males with a mesomorphic body type, and the minimum value was 53.4 mm in endomorphic females. The maximum changes were found in males with an endomorphic body type; it decreased by 14.6%. The change in the position of the lateral point of the outer edge of the middle part of the right thyroid lobe in the frontal plane, depending on the body type of the examined person at different positions of the head and neck, is shown in Table 2.

The distance (frontal plane) from the center of the jugular notch of the sternum to the outer edge of the lateral point of the middle part of the left thyroid lobe was determined, considering the gender and body type of the examined. When analyzing the distance from the center of the jugular notch to the outer edge of the lateral point of the middle part of the left thyroid lobe, taking into account the gender and body type of the examined person, it was found that in the average anatomical position, the maximum value of the distance studied in ectomorphic men is 76.7 mm; the smallest one is in women with an endomorphic body type (55.8 mm).

With a neck bent, the value of the studied variable decreases in all groups. The maximum value of the distance (frontal plane) from the center of the jugular notch of the sternum to the outer edge of the lateral point of the middle part of the left thyroid lobe was found in the group of examined males with an endomorphic body type; it was 73,03 mm. The minimum one was found in endomorphic women; it was 42.5 mm. The maximum changes were observed in the examined male volunteers with an ectomorphic body type; it decreased by 42.1%.

Neck extension has a multidirectional effect on the value of the studied variable: in the examined men with an ectomorphic body type, the distance from the center of the jugular notch of the sternum to the outer edge of the lateral point of the middle part of the left thyroid lobe decreased by 6.6%, while in the other groups, it increased. The maximum values of this variable were observed in male endomorphs (89.9 mm), and the minimum values were observed in women with an endomorphic body type (69.6 mm). The greatest changes were observed in the examined men with an endomorphic physique; it increased by 27.1%.

Rotating a head to the left has a multidirectional effect on the distance (frontal plane) from the center of the jugular notch of the sternum to the outer edge of the lateral point of the middle part of the left thyroid lobe. In both males and females with an ectomorphic body type, women with a mesomorphic body type, as well as in women with an endomorphic body type, there was a decrease in the distance. The highest value of the studied variable was found in the examined male endomorphs (79.6 mm), and the lowest one was found in female endomorphs (50.9 mm). The maximum change was a decrease of 17.1%; it was recorded in the examined males with an ectomorphic body type. In the groups of examined male mesomorphs and endomorphs, the study distance increased by 5.5% and 12.6%, respectively.

Rotating a head to the right has a multidirectional effect on the value of the variable under study. In males with mesomorphic and endomorphic body types, an increase in the distance in the frontal plane from the center of the jugular notch of the sternum to the outer edge of the lateral point of the middle part of the left thyroid lobe was noted by 4.1% and 12.6%, respectively, in all other groups, its decreased. The maximum value of the studied variable was obtained in men with an endomorphic body type; it was 79.6 mm. The minimum one was found in endomorphic women; it was 51.9 mm. The greatest changes were found in the examined men with an ectomorphic body type; it decreased by 18.4%. The change in the location of the lateral point of the outer edge of the middle part of the left thyroid lobe in the frontal plane, depending on the body type of the examined person at different positions of the head and neck, is shown in Table 3.

A comparative analysis of changes in the location of the lateral point of the outer edge of the middle part of the thyroid lobes in the frontal plane relative to the jugular notch, depending on the sex and body type of the examined at different positions of the head and neck, determined that the maximum value of the studied variable increases with neck extension. The change in the location of the lateral point of the outer edge of the middle part of the right and left thyroid lobes in the frontal plane, depending on the sex and body type of the examined at different positions of the head and neck, is shown in Figure 3.

The distance in the frontal plane from the center of the jugular notch of the sternum to the superior pole of the right thyroid lobe was determined, considering the gender and body type of the examined. When analyzing the distance from the center of the jugular notch to the superior pole of the right thyroid lobe, considering the gender and body type of the examined person, it was found that in the average anatomical position, the maximum value of the studied distance was noted in the examined women with an endomorphic body type (108.8 mm); the smallest one was noted in the group of endomorphic females (83.5 mm).

When a head is bent, the value of the studied variable decreases in all groups. Its maximum value was found in the group of males with an endomorphic body type (93.4 mm); the minimum value was found in endomorphic females (58.6 mm). The maximum change was found in the examined men with an ectomorphic body type; it decreased by 38.9%.

Head extension increases the value of the studied variable in all study groups, except for the group of males with an endomorphic body type, in which this value decreases by 4.42%. Its maximum values were found in males with an ectomorphic body type (120.6 mm); its minimum values were found in endomorphic females (90.2 mm). The greatest changes were recorded in the examined male mesomorphs; it increased by 12.6%.

Rotating a head to the left in all the examined groups leads to a decrease in the distance in the frontal plane from the center of the jugular notch of the sternum to the superior pole of the right thyroid lobe, except for the group of women with a mesomorphic body type (there was a 1.1% increase). The highest value of the studied variable was found in endomorphic males (108.6 mm), and the minimum value was found in the group of females with an endomorphic body type (75.6 mm). The maximum changes were recorded in the group of men with ectomorphic physique; it decreased by 7.3%.

Rotating a head to the right in all the examined groups leads to a decrease in the studied variable, except for mesomorphic females (there was a 1.08% increase). In all other groups, it decreased. The maximum value of the studied value was found in males with an endomorphic body type (106.3 mm); the minimum one was found in females with an endomorphic body type (75.8 mm). The maximum changes in the studied distance occurred in male ectomorphs; it decreased by 8.3%. The change in the location of the superior pole of the right thyroid lobe in the frontal plane, depending on the body type examined at different positions of the head and neck, is shown in Table 4.

Distance in the frontal plane from the center of the jugular notch of the sternum to the superior pole of the left thyroid lobe was determined, considering the gender and body type of the examined. When analyzing the distance from the center of the jugular notch to the superior pole of the left thyroid lobe, taking into account the gender and body type of the examined, it was found that in the average anatomical position, the maximum value of the studied distance occurred in ectomorphic men (105.9 mm), the smallest one occurred in the group of women with an endomorphic body type (81 mm).

When a head is bent, the value of this variable decreases in all groups. The maximum value of the studied variable was found in the group of men with a mesomorphic physique (71 mm); the minimum one was found in endomorphic women (59.6 mm). The maximum change was found in the examined women with mesomorphic physiques; it decreased by 37.8%.

Extension increases the distance in the frontal plane from the center of the jugular notch of the sternum to the superior pole of the left thyroid lobe in all study groups except for the group of mesomorphic women (there was a 1% decrease). The maximum values of the variable were found in the examined men with an ectomorphic physique (119.3 mm); the minimum one was found in endomorphic women (87.4 mm). The greatest changes were recorded in men with the ectomorphic body type; it increased by 13.3%.

Rotating a head to the left in all the studied groups leads to a decrease in the studied variable, with the exception of the group of males with a mesomorphic physique, in which an increase in this distance by 2% was detected. The highest value of this variable was found in mesomorphic males (100.6 mm); its minimum value was found in endomorphic females (80.1 mm). The maximum changes were recorded in the group of examined males with an ectomorphic physique; it decreased by 9.5%.

Rotating a head to the right in all the examined groups leads to a decrease in the studied variable, except for the group of men with the mesomorphic body type, in which an increase in this distance by 2% was detected. The maximum value of the studied variable occurred in the examined mesomorphic men (100.7 mm); the minimum one occurred in women with an endomorphic build (79.4 mm). The maximum changes were found in male ectomorphs; it decreased by 8.6%. The change in the location of the superior pole of the right thyroid lobe in the frontal plane, depending on the body type examined at different positions of the head and neck, is shown in Table 5.

A comparative analysis of changes in the position of the superior poles of the thyroid lobes in the frontal plane relative to the jugular notch of the sternum, depending on the sex and body type of the examined at different positions of the head and neck, determined that the maximum value of the studied variable increases with neck extension. The change in the location of the superior pole of the right and left thyroid lobes in the frontal plane, depending on the sex and body type of the examined at different positions of the head and neck, is shown in Figure 4.

## 4. Discussion

Modern anatomy studies the human body in a strictly defined position. There are not enough research works to determine changes in the relationship between organs and anatomical formations of the anterior neck during rotations and tilts of the head. At the same time, such information is of great practical importance.

The results obtained in the study allow us to state a significant displacement of the thyroid departments when the position of the head depending on gender and body type is changed. The knowledge gained about the intravital positional anatomy of the middle lobe and superior poles of the thyroid gland is not only theoretically important but also is of great practical momentousness. The obtained information will make it possible to personalize and optimize the approach to surgical treatment of patients with thyroid diseases, namely to facilitate surgical intervention by improving the basic parameters of surgical access [1,11].

The study of the intravital positional anatomy of the neck is relevant not only for surgeons who perform surgical interventions on the thyroid gland but also for specialists in other specialties such as neurosurgery, operative ophthalmology, vascular surgery, maxillofacial surgery, operative otorhinolaryngology, anesthesiology, military field surgery, plastic surgery, etc. In many works, authors describe the results of the analysis of the study of the positional anatomy of organs and structures of the neck using various instrumental methods [12]. Plastic and maxillofacial surgeons report changes in the position of the vessels of the neck when turning the head, in particular, the mutual displacement of the internal jugular vein and carotid artery [13], displacement of the mandibular branch of the facial nerve depending on the position of the head and neck [14]. Some of these studies are aimed at reducing the anesthetic risks during surgical interventions. For example, the authors report on choosing the optimal position of the head and neck to prevent aspiration of gastric contents during anesthesia [15]. Other studies indicate that neck extension can improve the visualization of objects during microlaryngoscopy [16,17]. Neurosurgeons describe that the different position of the head and neck during anesthesia changes the values of intracranial pressure [18].

Thus, the study of the intravital positional anatomy of the neck, particularly the thyroid gland, is an important subject of study of modern clinical anatomy and applied surgery.

The study focuses on healthy individuals, which may limit the applicability of the findings to patients with thyroid disorders or other neck-related conditions, including COVID-19 [19].

## 5. Conclusions

We found that the distance from the center of the jugular notch of the sternum to the lateral points of the outer edges of the middle parts of each of the thyroid lobes decreases with neck flexion in all the examined groups. This parameter changes most significantly in men with ectomorphic physiques (by 42.8% on the right and by 33.3% on the left). When the neck is extended in male ectomorphs, the distance from the center of the jugular notch of the sternum to the lateral points of the outer edge of the middle thyroid sections decreases by 5.2% on the right and 6.6% on the left. In the remaining groups of examined volunteers, an increase in this value was found, while the maximum increase in the parameter occurred in men with an endomorphic physique by 22% on the right and 27.1% on the left.

When the neck is bent, the distance from the center of the jugular notch of the sternum to the superior poles of the thyroid gland decreases in all groups examined. This distance changes most significantly in male ectomorphs on the right (by 38.9%) and in women with a mesomorphic body type on the left (by 37.8%). When the neck is extended, the distance from the center of the jugular notch of the sternum to the superior poles of the thyroid gland increases in all groups examined. The most significant changes were observed in men with a mesomorphic body type by 12.6% and in men with an ectomorphic body type by 13.3%.

### 5.1. Implications

Neck flexion reduces the distance from the center of the jugular notch of the sternum to the superior poles and lateral points of the outer edge of the middle parts of the thyroid lobes in all study groups by 78.4 mm to the maximum in endomorphic men and 44.1 mm to the minimum in ectomorphic men.Neck extension increases the distance from the center of the jugular notch to the superior pole and lateral points of the outer edge of the middle parts of the thyroid lobes by 100.1 mm maximum in endomorphic males and 73.8 mm minimum in ectomorphic males.The distance from the reference point of the sternum notch to the lateral points of the outer edge of the middle parts of the thyroid lobes decreases as much as possible when bending the neck in ectomorph males (by 42.1% on the left and 42.8% on the right). This distance is maximized when the neck is extended in male endomorphs on the right by 22% and on the left by 27.1%.The maximum distance from the reference point of the sternum notch to the superior pole of the thyroid gland decreases with neck flexion in ectomorph males by 38.9% on the right and mesomorph females by 37.8% on the left. Most of all, this distance increases in male ectomorphs on the right and left by 13.3%.Head rotations lead to a decrease in the distance from the center of the jugular notch of the sternum to the superior pole of the right and left thyroid lobes. When the head was rotated to the right, the maximum changes were observed in male ectomorphs—a decrease of 8.3% on the right and 8.6% on the left. When rotating the head to the left, the maximum changes were also recorded in the group of male-ectomorphs—a decrease of 9.5% on the left and 7.3% on the right.

### 5.2. Limitations

While this study includes 100 participants, the sample size might be insufficient to draw definitive conclusions, especially when considering the different body types and gender-specific variations.

## 6. Patents

Trunin, E.M.; Tatarkin, V.V.; Chernyshev, M.D.; Brovin, D.A.; Movchan, K.N.; Vasiliev, Yu. L.; Otochkin, V.V.; Bakunov, A.M.; Resnyanskaya, E.; Babaytseva, A.E.; Kulinich, V.A. Method for performing surgical operations on the skin of the patient. the thyroid gland. Patent for invention No. 2798713; Bul. No. 18, 23 June 2023 [7].

Trunin, E.M.; Tatarkin, V.V.; Chernyshev, M.D.; Brovin, D.A.; Shulga, V.P.; Bakunov, A.M.; Mustafina, E.A.; Shakhabadinov, V.Ya.; Eremina, A.A.; Podezhikh, S.; Rud, V.Yu.; Loseva, A.; Serebryakov, E.A.; Tyumenev, R.R. Device for fixing the patient’s head when performing magnetic resonance imaging of the neck. Patent for utility model No. 217872; Bul. No. 12, 21 April 2023 [10].

Trunin E.M.; Tatarkin V.V.; Shchegolev A.I.; Timokhov G.V.; Petrov. S.V.; Stetsik E.O.; Nechkin D.K.; Chernyshev M.D.; Vasiliev Y.L.; Bakunov A.M.; Alekseeva D.S.; Andreeva A.N.; Tolgsky M.V.; Lavrentieva A.N. Decision-making assistant in choosing the optimal personalized minimally invasive access to the thyroid gland. Certificate of state registration of a computer program No. 2024663047, 3 June 2024 [11].

## Figures and Tables

**Figure 1 medicina-60-01520-f001:**
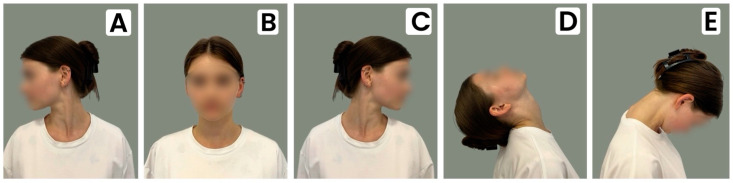
(**A**)—head rotated to the right; (**B**)—standard (physiological) anatomical head and neck position; (**C**)—head rotated to the left; (**D**)—position of maximum extension; (**E**)—position of maximum flexion.

**Figure 2 medicina-60-01520-f002:**
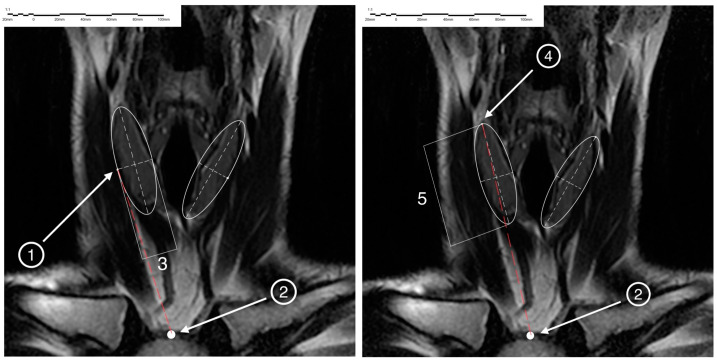
1—lateral point of the middle part of the thyroid lobe; 2—the jugular notch of the sternum; 3—MAX (for lateral point); 4—superior point of the superior pole; 5—MAX (for Superior pole).

**Figure 3 medicina-60-01520-f003:**
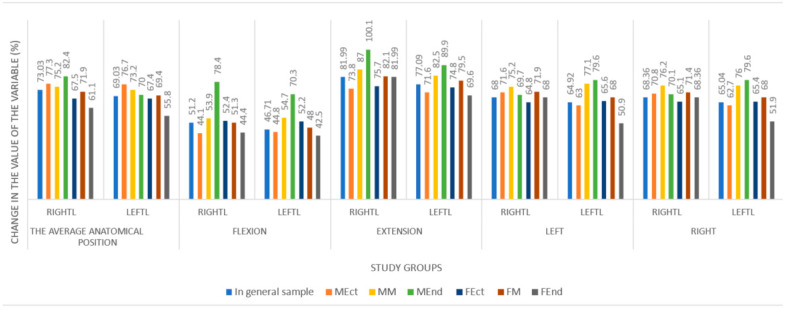
The changes in the location of the lateral point of the outer edge of the middle part of the right and left thyroid lobes in the frontal plane, depending on the body type examined at different positions of the head and neck.

**Figure 4 medicina-60-01520-f004:**
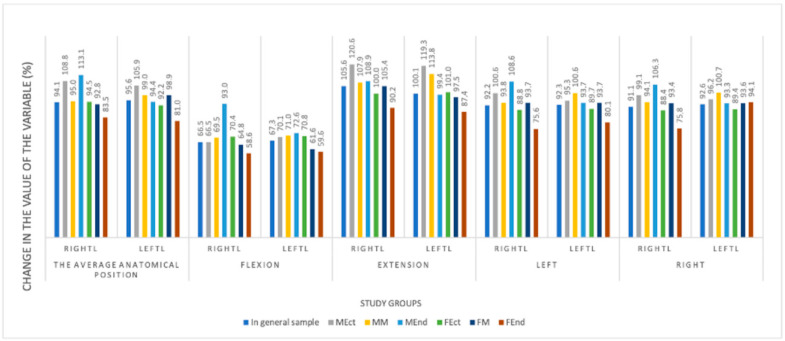
Changes in the location of the superior pole of the right and left thyroid lobes in the frontal plane depending on the body type examined at different positions of the head and neck.

**Table 1 medicina-60-01520-t001:** Scan parameters for performing thyroid magnetic resonance imaging.

Scan Plane	Parameters				
	FOV *	Slice Thickness	Spacing	Slices	TR *
3 Pl Loc *	25	5	5	15	-
Sag T2 *	25	4	1	23	3835
Cor T2 *	25	4	1.5	14	2200
Ax T2 *	25	4	1.5	20	2650

* 3 Pl Loc—3 Plane Localizer (Initial MR—sequence of any MR study obtained for planning main sequences); Sag T2—T2 weighted image in sagittal plane; Cor T—T2 weighted image in coronal plane; 2 Ax T2—T2 weighted image in axial plane; FOV—field of view (MR parameter meaning size of resulting MR tomogram image); TR—time of repetition (basic parameter of any MR-sequence meaning time in ms. between radio frequency pulses).

**Table 2 medicina-60-01520-t002:** Distance (frontal plane) from the center of the jugular notch of the sternum to the lateral point of the outer edge of the middle part of the right thyroid lobe, depending on the gender and body type of the examined (*p* < 0.05).

Title 1	Gender	Ectomorph	Mesomorph	Endomorph	In the General Sample
Average anatomical position Me (LQ; HQ) *	male	77.3 (74.2; 80.4)	75.2 (72.4; 78.0)	82.4 (79.3; 85.5)	73.0 (66.7; 78.0)
	female	67.5 (65.1; 69.9)	71.9 (69.0; 74.8)	61.1 (58.2; 64.0)	
Flexion Me (LQ; HQ)	male	44.1 (41.0; 47.2)	53.9 (51.1; 56.8)	78.4 (75.3; 81.5)	51.2 (45.1; 55.7)
	female	52.4 (50.0; 54.8)	51.3 (48.4; 54.2)	44.4 (41.5; 47.3)	
Extension Me (LQ; HQ)	male	73.8 (70.7; 76.9)	87.0 (84.1; 89.8)	100.1 (97.0; 103.2)	82.0 (74.0; 86.8)
	female	75.7 (73.3; 78.2)	82.1 (79.2; 85.0)	74.2 (71.3; 77.1)	
Left rotation Me (LQ; HQ)	male	71.6 (68.5; 74.7)	75.2 (72.3; 78.0)	69.7 (66.6; 72.8)	68.0 (64.0; 72.5)
	female	64.8 (62.3; 67.2)	71.9 (69.0; 74.8)	53.9 (51.0; 56.8)	
Right rotation Me (LQ; HQ)	male	70.8 (67.7; 73.9)	76.2 (73.4; 79.0)	70.1 (67.0; 73.2)	68.4 (63.8; 72.7)
	female	65.1 (62.7; 67.5)	71.4 (68.5; 74.3)	53.4 (50.5; 56.3)	

Me (LQ; HQ) *—median (lowest quantity; highest quantity).

**Table 3 medicina-60-01520-t003:** The distance (frontal plane) from the center of the jugular notch of the sternum to the outer edge of the middle part of the left thyroid lobe, considering the gender and body type of the examined (*p* < 0.05).

Title 1	Gender	Ectomorph	Mesomorph	Endomorph	In the General Sample
Average anatomical position Me (LQ; HQ) *	male	76.7 (73.2; 80.2)	73.2 (70.0; 76.4)	70.0 (66.5; 73.6)	69.0 (65.1; 73.1)
	female	67.4 (64.7; 70.2)	69.4 (66.1; 72.6)	55.8 (52.5; 59.1)	
Flexion Me (LQ; HQ)	male	44.8 (41.3; 48.3)	54.7 (51.6; 57.9)	70.3 (66.7; 73.8)	46.7 (44.0; 59.0)
	female	52.2 (49.5; 55.0)	48.0 (44.7; 51.2)	42.5 (39.2; 45.8)	
Extension Me (LQ; HQ)	male	71.6 (68.1; 75.1)	82.5 (79.3; 85.7)	89.9 (86.4; 93.4)	77.1 (71.6; 85.6)
	female	74.8 (72.1; 77.6)	79.5 (76.2; 82.8)	69.6 (66.4; 72.9)	
Left rotation Me (LQ; HQ)	male	63.0 (59.5; 66.5)	77.1 (73.9; 80.3)	79.6 (76.1; 83.1)	64.9 (58.9; 78.8)
	female	65.6 (62.9; 68.4)	68.0 (64.7; 71.2)	50.9 (47.7; 54.2)	
Right rotation Me (LQ; HQ)	male	62.7 (59.2; 66.2)	76.0 (72.8; 79.2)	79.6 (76.1; 83.1)	65.0 (58.9; 77.3)
	female	65.4 (62.7; 68.2)	68.0 (64.7; 71.3)	51.9 (48.6; 55.2)	

Me (LQ; HQ) *—median (lowest quantity; highest quantity).

**Table 4 medicina-60-01520-t004:** Distance (frontal plane) from the center of the jugular notch of the sternum to the superior pole of the right thyroid lobe, considering the gender and body type of the examined (*p* < 0.05).

Title 1	Gender	Ectomorph	Mesomorph	Endomorph	In the General Sample
Average anatomical position Me (LQ; HQ) *	Male	108.8 (104.5; 113.1)	95.0 (91.1; 98.9)	113.1 (108.8; 117.3)	94.1 (88.6; 109.8)
	Female	94.5 (91.2; 97.9)	92.8 (88.8; 96.8)	83.5 (79.5; 87.5)	
Flexion Me (LQ; HQ)	Male	66.5 (62.2; 70.8)	69.5 (65.6; 73.4)	93.4 (89.1; 97.7)	66.5 (61.9; 71.1)
	Female	70.4 (67.1; 73.8)	64.8 (60.8; 68.8)	58.6 (54.6; 62.6)	
Extension Me (LQ; HQ)	Male	120.6 (116.3; 124.9)	107.9 (104.0; 111.8)	108.9 (104.7; 113.2)	105.6 (96.2; 113.6)
	Female	100.0 (96.6; 103.3)	105.4 (101.3; 109.4)	90.2 (86.2; 94.2)	
Left rotation Me (LQ; HQ)	Male	100.6 (96.3; 104.9)	93.8 (89.9; 97.7)	108.6 (104.3; 112.9)	92.2 (85.6; 103.4)
	Female	88.8 (85.4; 92.1)	93.7 (89.7; 97.7)	75.6 (71.6; 79.6)	
Right rotation Me (LQ; HQ)	Male	99.1 (94.8; 103.4)	94.1 (90.2; 98.0)	106.3 (102.0; 110.6)	91.1 (85.7; 102.1)
	Female	88.4 (85.1; 91.7)	93.4 (89.4; 97.4)	75.8 (71.8; 79.8)	

Me (LQ; HQ) *—median (lowest quantity; highest quantity).

**Table 5 medicina-60-01520-t005:** Distance (frontal plane) from the center of the jugular notch of the sternum to the superior pole of the left thyroid lobe, considering the gender and body type of the examined (*p* < 0.05).

Title 1	Gender	Ectomorph	Mesomorph	Endomorph	In the General Sample
Average anatomical position Me (LQ; HQ) *	Male	105.9 (102.3; 109.6)	99.0 (95.7; 102.3)	94.4 (90.8; 98.0)	95.6 (86.7; 103.1)
	Female	92.2 (89.3; 95.0)	98.9 (95.6; 102.3)	81.0 (77.7; 84.4)	
Flexion Me (LQ; HQ)	Male	70.1 (66.5; 73.8)	71.0 (67.7; 74.3)	72.6 (69.0; 76.2)	67.3 (62.0; 72.1)
	Female	70.8 (68.0; 73.6)	61.6 (58.2; 65.0)	59.6 (56.2; 62.9)	
Extension Me (LQ; HQ)	Male	119.3 (115.7; 123.0)	113.8 (110.5; 117.1)	99.4 (95.8; 103.1)	100.1 (95.0; 111.7)
	Female	101.0 (98.1; 103.8)	97.5 (94.2; 100.9)	87.4 (84.0; 90.8)	
Left rotation Me (LQ; HQ)	Male	95.3 (91.7; 98.9)	100.6 (97.3; 103.9)	93.7 (90.1; 97.3)	92.3 (86.9; 97.9)
	Female	89.7 (86.9; 92.6)	93.7 (90.3; 97.1)	80.1 (76.7; 83.4)	
Right rotation Me (LQ; HQ)	Male	96.2 (92.6; 99.8)	100.7 (97.4; 103.9)	93.3 (89.7; 97.0)	92.6 (87.4; 98.4)
	Female	89.4 (86.5; 92.2)	93.6 (90.2; 97.0)	79.4 (76.0; 82.8)	

Me (LQ; HQ) *—median (lowest quantity; highest quantity).

## Data Availability

The raw data supporting the conclusions of this article will be made available by the authors on request.

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
