# Peer review of "Intravital Position Study of the Clinical Anatomy of the Middle Lobe and Superior Poles of the Thyroid Gland"

_medicina, 2024, doi:10.3390/medicina60091520_

Round 1
Reviewer 1 Report
Comments and Suggestions for Authors
Abstract: p values should be written. Significant parameters should be mentioned.
Keywords should be written from MESH on Demand tool.
L30-35, 44-45, 51-55: Introduction: Missing references should be added. Abbreviations should be explained before. Additionally, abbreviations should be checked in overall the text and check from MEDLINE.
L60-62: It needs editing
Figure 2: Scale bar, directions should be added
L116-120: Units should be written.
L147-150, 164-166: It needs editing. It is not clear.
Figures 3-4: Image quality should be increased. Significant values should be indicated on figures.
L377, L382: ‘In many works, authors describe results of..’ , ‘Some of these studies..’ It needs references
L378-382:’ Publications of plastic and maxillofacial surgeons report changes in the position of the vessels of the neck when turning the head, in particular, the mutual displacement of the internal jugular vein and carotid artery [12], displacement of the mandibular branch of the facial nerve depending on the position of the head and neck ‘It also effects the other vessels and organs? And what about the effects?
L359-391: Discussion is not enough in the current form. It should be divided into sub-sections according to the parameters
L391-392: Limitations part should be added, Suggestions part should be recommended.
L392-410: What are the contributions of these findings to the litaerature and clinical practice ? Please write the contributions to the literature and novelty of your findings replace for findings. what about the effects on surgery?
Tables 2-5: p values should be added.
References: doi numbers should be added, missing references in the text and recent studies should be added.
Comments on the Quality of English LanguageModerate editing of English language required.
Author Response
Dear Reviewer,
We appreciate the time you spent on our manuscript.
Let us answer your questions one by one
1. Abstract: p values ​​should be written. Significant parameters should be mentioned. – Corrections have been made on lines L20-21; L25-27
2. Keywords should be written from MESH on Demand tool – corrections have been made on line 30
3. L30-35, 44-45, 51-55: Introduction: Missing references should be added. Abbreviations should be explained before. Additionally, abbreviations should be checked in the overall text and checked from MEDLINE. – I did not find any abbreviations in the text, the citation is correct
4. Figure 2: Scale bar, directions should be added – we added a ruler, graph paper
5. L116-120: Units should be written. – corrections made on lines L118-119 6. Figures 3-4: Image quality should be increased. Significant values ​​should be indicated on figures. – done 7. L377, L382: ‘In many works, authors describe results of..’ , ‘Some of these studies..’ It needs references – link added, correction can be found on line 380 8. L378-382: ‘Publications of plastic and maxillofacial surgeons report changes in the position of the vessels of the neck when turning the head, in particular, the mutual displacement of the internal jugular vein and carotid artery [12], displacement of the mandibular branch of the facial nerve depending on the position of the head and neck ‘It also effects the other vessels and organs? And what about the effects? – only a description of the displacement of organs and vessels on lines 382-383, without a description of the influence.
9. L392-410: What are the contributions of these findings to the literature and clinical practice? Please write the contributions to the literature and novelty of your findings to replace for findings. what about the effects on surgery? Necessary comments are on lines 368-373
10. Tables 2-5: p values ​​should be added. - Tables 2-5 do not contain a comparative component. Only descriptive, median and quartiles. Accordingly, the p value was not determined in this form.
11Ю References: doi numbers should be added, missing references in the text and recent studies should be added. - so where data was found, we added it.
Reviewer 2 Report
Comments and Suggestions for Authors
The study is clinically important and highly relavant. I have the following major concerns related to this manuscript.
Introduction
Please state if there are any similar studies earlier in the literature; if yes or no, please state with a proper citation.
The rationale for the study should be revised with a more precise objective.
The following statement should be modified to make the objective of the study more precise.
“Our purpose was to gain information on the intravital positional anatomy of the superior poles and lateral areas of the middle parts of each of the thyroid lobes in both males (57 and females with different body shapes.”
Methods
The statement “Was used the data of anthropometry” should be paraphrased. The same sentence is in the abstract as well.
The study design should be mentioned.
The exclusion criteria should be added.
References should be provided for all methods mentioned. If it’s a new method, please justify why it was considered.
The company details should be provided for machines, scanners, and software.
Was consent from the study subjects taken? If yes, please state the same along with the ethical statement.
How many investigators involved in the study should be stated, and if more than one, how was the risk of bias avoided?
Divide the methods into sections for better understanding by the readers.
The statistical method used is unclear.
Results
Similar to methods, divide the results section into subsections.
Add the table legends for each table. Specify the units for the values given in each table.
Overall, the figures are clumsy; it would be better to use a different format to present the data. The figure should be modified for better clarity.
Discussion
The discussion should be rewritten in the order of findings. Each major finding should be supplemented with clinical significance.
Limitations and future directions should be added.
Comments on the Quality of English LanguageMinor editing of English language required.
Author Response
Dear Reviewer,
We appreciate the time you have spent on our manuscript.
Let me answer your questions in order
1. Please state if there are any similar studies earlier in the literature; if yes or no, please state with a proper citation. – не беда
2. The rationale for the study should be revised with a more precise objective. - corrections have been made on lines 58-60
3. The following statement should be modified to make the objective of the study more precise. - necessary corrections made
4. “Our purpose was to gain information on the intravital positional anatomy of the superior poles and lateral areas of the middle parts of each of the thyroid lobes in both males (57 and females with different body shapes.” - necessary correction on lines 58-60
5. The statement “Was the data of anthropometry used” should be paraphrased. The same sentence is in the abstract as well. - corrections made on lines 20 and 61
The study design should be mentioned. – corrections made on lines 62-L80.
6. The company details should be provided for machines, scanners, and software. – these data are on line 85, Table 1, line 143
7. Was consent from the study subjects taken? If yes, please state the same along with the ethical statement. – the decision of the ethical committee is written on lines 67-69
8. The statistical method used is unclear. – changes on lines 142-147
9. Add the table legends for each table. Specify the units for the values ​​given in each table. – corrections were made to the description of Table 2-5 – one decimal place, description added.
10. Overall, the figures are clumsy; it would be better to use a different format to present the data. The figure should be modified for better clarity. – corrections were made to the description of Table 2-5 – one decimal place
11. Limitations and future directions should be added. limitations were added on lines 393-395
Reviewer 3 Report
Comments and Suggestions for Authors
The article titled "Intravital Position Study of the Clinical Anatomy of the Middle Lobe and Superior Poles of the Thyroid Gland" explores the positional anatomy of the thyroid gland, particularly focusing on the middle lobe and superior poles. The research aims to provide insights into how the position of the thyroid gland changes during various neck movements. The study is based on data from MRI scans of 100 healthy participants of different body types. The primary challenge was to accurately capture the anatomical changes in the thyroid gland during different movements and across different body types and genders.
While the study includes 100 participants, the sample size might be insufficient to draw definitive conclusions, especially when considering the different body types and gender-specific variations. Please add this as a limitation of the study.
The study focuses on healthy individuals, which may limit the applicability of the findings to patients with thyroid disorders or other neck-related conditions, including COVID-19 infection. Please add the reference https://doi.org/10.47162%2FRJME.63.1.03.
The paper cites relevant literature, but additional references could strengthen the discussion, particularly in the context of minimally invasive thyroid surgery.
Author Response
Dear Reviewer,
We appreciate your time spent on our manuscript.
Let me answer your questions one by one
1. While the study includes 100 participants, the sample size might be insufficient to draw definitive conclusions, especially when considering the different body types and gender-specific variations. Please add this as a limitation of the study. Limitations are added on lines 393-395
2. The study focuses on healthy individuals, which may limit the applicability of the findings to patients with thyroid disorders or other neck-related conditions, including COVID-19 infection. Please add the reference https://doi.org/10.47162%2FRJME.63.1.03.
corrections in the text on lines 393-395 and in the list of references 516-519
3. The paper cites relevant literature, but additional references could strengthen the discussion, particularly in the context of minimally invasive thyroid surgery. text correction on lines 380 and 396
Round 2
Reviewer 2 Report
Comments and Suggestions for Authors
- The revised manuscript improved significantly. However, I have a few other comments for authors to consider.
- In the introduction in the following sentence, the number of males can be removed. This information should be mentioned in the methods.
“Our purpose was to gain information on the intravital positional anatomy of the superior poles and lateral areas of the middle parts of each of the thyroid lobes in both males (57 male and female with different body shapes).”
- In the discussion, the phrase “publications of plastic and maxillofacial surgeons” and similarly in other sentences, the “publications” should be removed. Instead, the study describes, or researchers reported, etc.
- In discussion, authors should justify all their results with relevant arguments in the literature. And emphasize its significance in a broader way.
- Please cite all relevant literature in both the introduction and discussion sections.
Minor editing of English language required.
Author Response
Hello, dear reviewer! Thank you for your valuable comments, which helped us to qualitatively improve the level of work.
Comment 1
In the introduction in the following sentence, the number of males can be removed. This information should be mentioned in the methods. “Our purpose was to gain information on the intravital positional anatomy of the superior poles and lateral areas of the middle parts of each of the thyroid lobes in both males (57 male and female with different body shapes).”
Response 1: improved, it was a misprint. (L59-60)
Comment 2
In the discussion, the phrase “publications of plastic and maxillofacial surgeons” and similarly in other sentences, the “publications” should be removed. Instead, the study describes, or researchers reported, etc.
Response 2: Improved, according to the suggestions described above. (L380-383; 387-389)
Comment 3: in discussion, authors should justify all their results with relevant arguments in the literature. And emphasize its significance in a broader way.
Response 3: without any changes, relevant arguments in the literature are already written.
Comment 4: please cite all relevant literature in both the introduction and discussion sections. Response 4: without any change, all relevant literature is already cited.